# Arbuscular Mycorrhiza Support Plant Sulfur Supply through Organosulfur Mobilizing Bacteria in the Hypho- and Rhizosphere

**DOI:** 10.3390/plants11223050

**Published:** 2022-11-11

**Authors:** Jacinta Gahan, Orla O’Sullivan, Paul D. Cotter, Achim Schmalenberger

**Affiliations:** 1Department of Biological Sciences, School of Natural Sciences, University of Limerick, V94 T9PX Limerick, Ireland; 2Teagasc Food Research Centre, Moorepark, Fermoy, and APC Microbiome Ireland, P61 C996 Cork, Ireland

**Keywords:** arylsulfonate, sulfate-ester, ^34^S stable isotope, hyphosphere, *asfA*, *Agrostis stolonifera*, *Plantago lanceolata*, arbuscular mycorrhiza

## Abstract

This study aimed to elucidate the role of bacteria colonising mycorrhizal hyphae in organically bound sulfur mobilisation, the dominant soil sulfur source that is not directly plant available. The effect of an intact mycorrhizal symbiosis with access to stable isotope organo-^34^S enriched soils encased in 35 µm mesh cores was tested in microcosms with *Agrostis stolonifera* and *Plantago lanceolata*. Hyphae and associated soil were sampled from static mesh cores with mycorrhizal ingrowth and rotating mesh cores that exclude mycorrhizal ingrowth as well as corresponding rhizosphere soil, while plant shoots were analysed for ^34^S uptake. Static cores increased uptake of ^34^S at early stages of plant growth when sulfur demand appeared to be high and harboured significantly larger populations of sulfonate mobilising bacteria. Bacterial and fungal communities were significantly different in the hyphospheres of static cores when compared to rotating cores, not associated with plant hosts. Shifts in bacterial and fungal communities occurred not only in rotated cores but also in the rhizosphere. Arylsulfatase activity was significantly higher in the rhizosphere when cores stayed static, while *atsA* and *asfA* gene diversity was distinct in the microcosms with static and rotating cores. This study demonstrated that AM symbioses can promote organo-S mobilization and plant uptake through interactions with hyphospheric bacteria, enabling AM fungal ingrowth into static cores creating a positive feedback-loop, detectable in the microbial rhizosphere communities.

## 1. Introduction

The essential macro-nutrient sulfur (S) is increasingly limiting crop yield and quality as a result of reduced atmospheric deposition while high yielding crop varieties rapidly deplete soil S stocks [1]. These factors are exacerbated by the fact that S present in soil is approximately 95% organically bound as sulfate-esters and sulfonates [2,3] and not directly available to plants which rely upon soil microbial populations for organo-S mineralisation [4].

The ability to mineralise sulfate-esters is common for many bacteria and saprotrophic fungi in soil [5,6]. Genes encoding for arylsulfatases that cleave the O-S bond appear to share common conserved regions. One of these arylsulfatase genes was described in *Pseudomonas aeroginosa* as *atsA* [5,6] and has been most comprehensively studied for various *Pseudomonas* species [7,8]. More recently *atsA* has been established as a molecular marker for arylsulfatase gene diversity studies [9]. The ability of soil bacteria to desulfurize a large variety of sulfonates is based on a multicomponent mono-oxygenase enzyme complex that includes the *ssu* and *asf* gene clusters [10]. For aliphatic sulfonate desulfurization the *ssuD* gene has been developed as a molecular marker [9], while for aromatic sulfonate desulfurization, *asfA* is an established marker [11,12,13,14,15]. The *asfA* gene involved in aromatic sulfonate mobilization has been identified in the Beta-Proteobacteria *Variovorax, Polaromonas*, *Hydrogenophaga*, *Cupriavidus*, *Burkholderia* (*Paraburkholderia*) and *Acidovorax,* the Actinobacteria *Rhodococcus* and *Williamsia*, and the Gamma-Proteobacteria; *Pseudomonas* and *Stenotrophomonas* [11,12,13,14,15].

Soil organosulfur transformation (S cycling) involves complex interactions between free living and symbiotic rhizospheric microbial populations. Arbuscular mycorrhizal (AM) fungi are part of these rhizospheric microbial populations and form an endosymbiosis with 77% of angiosperms, 45% of gymnosperms and 52% of fern and lycopod [16]. Their characteristic structure, the arbuscule, is extensively branched and acts as an efficient site for metabolite exchange with the host plant [17]. Furthermore, their extraradical hyphae (ERH) play an important role in nutrient acquisition via extensive networks of microscopic hyphae that access a volume of soil orders of magnitude greater than plant roots [18]. Studies have established for some time now that the presence of AM fungi enhances S uptake for maize, clover [19] and tomato [20] but to date little is known about the actors or the exact processes that facilitate this increased uptake. In vitro studies with transformed carrot roots demonstrated the uptake of reduced forms of S (e.g., cysteine, methionine) in the presence of the AM fungus *Rhizophagus irregularis* (formerly known as *Glomus intraradices*) [21]. Rates of uptake and transfer of reduced S were comparable to that of SO_4_^2−^ when the latter was limited. However, it is currently not clear whether this form of uptake is important in vivo. Uptake of SO_4_^2−^ in the rhizosphere leads to a zone of depletion analogous to that observed for phosphorus [22]. The AM fungal ERH could extend past this zone of SO_4_^2−^ depletion and play an important role in provision of S under conditions of limitation [4]. Additionally, more recent investigations have revealed that AM fungi influence the expression of plant sulfate transporters and as a consequence improve S uptake [23].

AM symbiosis alters the rhizospheric microbial community composition by modifying the biochemical composition of root exudates [24,25]. Additionally, translocation of energy rich C compounds to the extended soil environment via their extensive ERH networks provides an important niche for functional interactions [26]. Diverse soil microbial communities are essential for soil fertility and plant vitality [27,28]. Likewise, AM hyphae in a native grassland ecosystem have been shown to host a large community of sulfonate mobilising bacteria with potential to improve plant S supply [15]. Moreover, it has been shown that the addition of morpholine ethane sulfonic acid to soil stimulated sulfonate mobilising bacteria and their metabolites enhanced ERH growth of *R. irregularis* (then *G. intraradices* [29]). This is important for maximising S uptake as enhanced hyphal growth stemming from sulfonate mobilising bacterial metabolites may further stimulate the proliferation of this community in a potential positive feedback loop. Despite the above reported findings, a direct link between the bacterial organo-S mobilizing activity in the hyphosphere of AM fungi and the supply of S of organo-S origin to plants remains elusive.

The hypothesis of this study was that (*i*) ingrowth of symbiotic AM hyphae into organo-^34^S enriched mesh cores would increase plant S uptake and that (*ii*) the hinderance of AM hyphae ingrowth would affect the organo-S mobilising microbes. Since the latter facilitate the organo-S mineralization, rotating the mesh cores containing organo-^34^S will affect the uptake of newly mineralised S through AM hyphae. The hypothesized associated activity of organo-S mineralization and transport is important to overcome plant S limitations and to reduce over-dependencies on synthetic fertilizer use.

## 2. Results

### 2.1. Plant Uptake of ^34^S from Organo-^34^S Enriched Soil Microcosms

For *P. lanceolata,* total sulfur uptake was significantly increased (*p* < 0.05) for the static cores over the rotating cores at 3 months (0.23 ± 0.006, vs. 0.20 ± 0.02% dry matter (DM)), but not at 6 months. Likewise, *A. stolonifera*, sulfur uptake increased significantly for the static cores for 3 and 6 months but not for month 9 (0.14 ± 0.01, 0.23 ± 0.03 and 0.23 ± 0.04% for rotating cores and 0.16 ± 0.01, 0.31 ± 0.03 and 0.23 ± 0.04% DM for the static cores, respectively).

Very similar results were obtained for the ^34^S uptake. The δ^34^S_V-CDT_ (‰) values were significantly higher for the static cores for *Agrostis* and *Plantago* after 3 months of incubation when compared to the rotating core experiment (Figure 1). After 6 months this was only the case for *Agrostis* and after 9 and 12 months, no significant difference in δ^34^S_V-CDT_ was detected in the *Agrostis* plants (Figure 1 and Appendix A). The δ^34^S_V-CDT_ (‰) values were relatively small at this time point (30 compared to 750 δ^34^S_V-CDT_ upon original analysis) which may explain the absence of a significant effect.

### 2.2. S K-Edge X-ray Absorption near Edge Spectroscopy

XANES analysis was undertaken to identify S oxidation states and revealed highly similar spectra for the static and rotating treatments with *A. stolonifera* with reduced (sulfide, thiols), intermediate (sulfoxides and sulfonates) and oxidised (sulfate esters) S species peaks. The only observed difference between the static and rotating cores was a slightly more pronounced sulfide/thiol peak for the static treatment (Appendix A). The photon energy of reduced thiols (2474.4 eV), intermediate sulfonates (2480.2–2480.4 eV) and oxidised sulfate esters (2481.6 eV) were identified in a previous study [30] and correspond to the three main peaks in the present study. The photon energy for the corresponding S species in this study appear to be slightly larger due to a larger step size increment utilised in this study (0.5 eV) in comparison to the way the reference values were obtained (0.2 eV).

### 2.3. AM Colonisation

The extent of root intracellular colonisation with characteristic AM structures was assessed for both *A. stolonifera* and *P. lanceolata* (Table 1). Significantly higher arbuscular, vesicular and hyphal colonization (AC, VC and HC) values were observed for static over rotating core treatments (*p* < 0.05) that represented a 59–62% increase. Likewise, rates of AC were higher for *Agrostis* (55%) and *Plantago* (58–61%) when compared to VC rates.

### 2.4. Quantification of Cultivable Bacteria

For both plants, cultivable heterotrophic (R2A) and sulfonate mobilising (MM2TS) bacterial communities were more abundant in the microcosms with the static treatment. This was the case for the static cores as well as the rhizosphere (roots outside the cores) when compared to the respective rotating treatment (Table 1). Heterotrophs had a six-fold higher abundance in static *A. stolonifera* microcosms (cores and rhizosphere, *p* < 0.05), while for *P. lanceolata* the increase in abundance was six- (cores) and twelve-fold (rhizosphere) for static over rotating treatments (*p* < 0.05). Additionally, for *A. stolonifera* the polymeric sulfonate mobilising (MM2LS) bacterial community was more abundant in the static treatment (rhizosphere and cores) over the respective rotating treatment (twenty-fold; *p* < 0.05). For *P. lanceolata,* the polymeric sulfonate mobilising bacterial abundance was not significantly different among the cores but was ten-fold higher in the static rhizosphere when compared to the rotating treatment (*p* < 0.05; Table 1).

Arylsulfonate mobiliser abundances varied the greatest (10^4^–10^7^ MPN g^−1^), while the polymeric sulfonate mobilising bacteria were lower in abundance than the arylsulfonate mobilizers (10^3^–10^5^ MPN g^−1^; Table 1). The heterotrophic bacterial abundance was similar in the cores and rhizospheres for both plants (*p* > 0.05) for the rotating and the static treatment, respectively. In contrast, aromatic sulfonate mobilisers were highest in the static cores for both plants (Table 1).

### 2.5. Fingerprinting Based Community Analysis

Detrended correspondence analysis (DCA) of the Denaturing Gradient Gel Electrophoresis (DGGE) fingerprints was used in conjunction with Monte Carlo permutation testing to identify differences in the beta diversity of bacterial and fungal communities for *A. stolonifera* and *P. lanceolata* in the cores and the rhizosphere (Figure 2, Figure 3 and Figure 4). For the bacterial communities, the rotating and static cores and their respective rhizospheres contained significantly different bacterial communities (*p* < 0.05; Figure 2). While the core samples of *Agrostis*, were separated well on the first axis, the rhizosphere samples were separated on the second axis (Figure 2A). This was the other way round for the bacterial communities in the *Plantago* microcosms (Figure 2B). The AM fungal communities differed also significantly for the cores and rhizospheres for both plants (*p* < 0.05; Figure 3). In the *Agrostis* microcosms, the core samples were separated on the second axis while the rhizosphere samples were separated on the first axis (Figure 3A). For *Plantago*, the separation of cores and rhizospheres was the other way round (Figure 3B). The saprophytic fungal communities (Figure 4, based on ITS fingerprints) differed significantly between the cores and rhizospheres for *Plantago* where the rhizosphere samples were separated on the first axis, while the core samples were separated on the second axis (*p* < 0.05; Figure 4B). For *Agrostis* the rotating and static cores were significantly different and also differed significantly to the rhizospheres, but no significant difference was detected between the rhizosphere samples of static and rotating core microcosms (Figure 4A).

### 2.6. Arylsulfatase Activity

Sulfate ester mobilising activity was measured in the hyphosphere (cores) and rhizosphere of the static (mycorrhizal) and rotating (severed hyphae) treatments. For both plants, arylsulfatase activity was higher for the rhizosphere than the cores and, additionally, for the rhizospheres from the static over the rotating treatment (*p* < 0.05). For *A. stolonifera,* 19.1 (±1.09) and 11.8 (±1.90) µg of *p*-nitrophenol was released in the rhizosphere from the static and rotating treatments, respectively. For the static and rotating cores this was lower at 5.5 (±1.22) and 8 (±2.23) µg, respectively and was not significantly different (*p* > 0.05). For *P. lanceolata,* 23.4 (±2.25) and 17 (±2.30) µg of *p*-nitrophenol was released from rhizospheres of the static and rotating treatments, respectively. This was significantly higher when compared to the static and rotating cores at 2.5 (±0.50) and 2.5 (±0.30) µg which were identical (*p* > 0.05) (Appendix A).

### 2.7. Diversity of asfA and atsA Gene

Screening of 200 *asfA* clones (50 each per treatment in *A. stolonifera* microcosms) revealed 43 unique OTUs in total; 12 in static cores, 10 in rotating cores, 12 in the rhizosphere of the static and 15 in rhizosphere of the rotating treatments. Only 2 OTUs occurred in all treatments. Both OTUs were the most abundant composing 23% and 12% of the sulfonate mobilising population analysed. Library coverage was calculated to be between 92 and 96%. 18 OTUs had more than three representatives and were subjected to Sanger sequencing. Initial identification of the phylogenetic affiliation was determined with NCBI’s BLAST [31]. All sulfonate mobilising bacterial marker gene fragments were found to belong to the phylum Proteobacteria (Table 2). Sequences (*asfA*) were translated and incorporated into a phylogenetic tree ([32] (Appendix A). The two OTUs present in all clone libraries (CS16 and CS37) clustered in a clade that contains the genus *Cupriavidus* (Appendix A).

Five OTUs associated with *Variovorax* and one OTU associated at the family level of the *Commamonadaceae* (Table 2) clustered within the *Variovorax* clade of the phylogenetic tree (Appendix A). Together, they represented 16% of the *asfA* clones. None of the *Variovorax* associated clones were found in the static mesh cores, while eight were found within the rotating cores and 23 clones were found in the plant rhizosphere. The *Burkholderia* clade harboured one OTU (CS12), while the *Polaromonas* clades contained 2 OTUs (RS4, RS6). OTU CS12 was found to be associated with the genus now identified as *Paraburkholderia*. Only recently, the genus *Burkholderia* has been split into two genera, with most of the environmental strains moved into the newly established genus of *Paraburkholderia* [33]. The clade that was associated with *Cupriavidus* contained 40% of the *asfA* clones that included CS37 and CS16. Notably, 29 of the 46 clones of CS16 were found on fungal hyphae in static mesh cores (presumptive mycorrhizal). One OTU was found to be closely associated with *Stenotrophomonas* (CR38; BLAST and phylogenetic tree) that was only found in the rotating mesh core.

Clone libraries of the sulfate-ester mobilising *atsA* gene amplicons (160 clones, 40 per treatment) contained 38 OTUs in total; 9 for static cores (5 overlapping), 14 for rotating cores (6 overlapping)*,* 15 for static rhizosphere (5 overlapping), and 13 for rotating rhizosphere (5 overlapping). Library coverage [34] was at or above 85%. OTUs with more than three representatives (15) were sequenced and phylogenetic affiliation established via BLAST (Table 3).

The dominating sulfate-ester mobilising bacteria in possession of the *atsA* marker gene were found to be associated with Acidobacteria (27.5%)*,* Verrucomicrobia (14%), Bacteroidetes (12.5%) and Rhodospirillales (order) (10%) (Table 3). One OTU, representing 4% of *atsA* sequences was associated with the eukaryotic phylum Rotifera. Differences in abundance were observed across the four treatments. Acidobacteria were more abundant in the rhizosphere, particularly in the rotating treatment. Rhodospirillales like clones were predominantly isolated from the hyphosphere (cores) and were present in both static and rotating treatments (Table 3).

### 2.8. NGS Based Bacterial Community Sequences

Using the Chao 1 index [35], the species diversity in the rotating and static cores (*A. stolonifera* as host plant) was not determined to be different at 4006 (±1624) and 4774 (±1242), respectively. The number of OTUs obtained from the rotating and static core samples were determined to be 3955 (±1644) and 4740 (±1246), respectively. The rotating core samples had a higher Simpson and Shannon index (0.97 and 7.4) than the static core samples (0.87 and 6.6, respectively) [36,37].

PCoA (un-weighted, Unifrac distance matrix) separated the rhizosphere and hyphosphere very clearly (Figure 5). While the rhizosphere treatments clustered together, rhizosphere and core samples were clearly separated on the first axis. A separation between the static and rotating cores (hyphosphere) was visible on the second and third axis (Figure 5).

Bacterial phyla that were most abundant in the rhizosphere treatments (with static and rotating cores) included the Acidobacteria (16–20%), Actinobacteria (7–9%), Planctomycetes (4%), Proteobacteria, (28–31%) and Verrucomicrobia (9–10%). Bacterial phyla that were most abundant in the hyphosphere treatments (within the cores, static or rotating) included the Acidobacteria (6–13%), Actinobacteria (2–10%), Bacteroidetes (2–4%), Firmicutes (3–5%), Planctomycetes (22–34%) and Proteobacteria (24–29%) (Appendix A). Of the 18 most abundant phyla, only the Planctomycetes and Gemmatimonadetes were significantly more abundant in the static than the rotating cores, while only the Acidobacteria were significantly more abundant in the rotating over the static cores (*p* < 0.05). Differences in relative phyla abundance was more widespread between the cores and the rhizospheres (rotating and static) with 11 of the 18 most abundant phyla showing significant differences in abundance (data not shown).

Among the 35 most abundant bacterial families, significant differences (*p* < 0.05) between the cores (hyphosphere, static, rotating) and the rhizospheres were detected (Table 4, marked in bold). Nine of the 35 most abundant families were significantly more abundant in the cores, most notably the Holophagaceae and Xanthomonadaceae. In contrast, 17 of the 35 most abundant families were significantly more abundant in the rhizospheres than in the cores and included the Acidobactertiaceae, Xanthobacteraceae and the DA101 group (Table 4). Significant differences between the static and rotating cores were rare. Only two of the 35 most abundant families were more abundant in the static (Gemmatimonadaceae, Sphingomonadaceae) and the rotating (DA111 group, Sorangiinae) cores, respectively (Table 4). Significant differences between the rhizospheres of static and rotating core microcosms could not be determined as several replicates of the static rhizosphere samples failed the quality controls of the sequencing run and were subsequently removed from the analysis.

## 3. Discussion

AM fungi are highly beneficial to plant hosts as they are involved in nutrient mobilisation as determined with organically bound labelled isotopes P and N [38,39,40]. Additionally, AM fungi have been shown to promote plant growth [41] and harbour abundant sulfonate mobilising bacterial communities [15]. While little evidence exists to suggest that AM fungi directly mobilise organically bound S, improved S uptake has been observed in their presence [21]. Indeed, ERH growth has been reported to be enhanced in the presence of organo-S where bacterial activity was suspected to be the driving force behind the growth rates of ERH [29]. The present study aimed to illustrate that the presence of an intact AM symbiosis increases uptake of organo-S via organo-S mobilising microbes and identified increased ^34^S uptake from organo-S in the first three months of growth of *Agrostis* and *Plantago*.

Plant S demand is dependent both on plant species and stage of development with increased demand observed during periods of vegetative growth and seed development [42]. When S demand is high, plant SO_4_^2−^ transporters are up-regulated for rapid uptake in the rhizosphere leading to a SO_4_^2−^ depletion zone [22]. In this zone, bacterial desulfurization of organo-S is induced via AM influenced rhizodeposition [3]. When S demand is lessened at later growth stages, it is no longer viable to expend carbon to stimulate proliferation of organo-S mobilisers and the AM role may be minimised. Findings from the present study appear to support this scenario, as ^34^S plant uptake from organo-^34^S enriched soil was increased in the microcosms with static cores after 3 months for *A. stolonifera* and *P. lanceolata* and for *A. stolonifera* also after 6 months. XANES analysis undertaken in the present study uncovered that sulfate-esters and sulfonates were both present in the soil. XANES analysis also revealed that the static cores may have contained more of the reduced S species, which suggests a higher S uptake and incorporation into fungal biomass in the static cores. Indeed, more abundant reduced S has been associated previously with fungal biomass [43]. Other studies have used a conventional soil fractionation approach to estimate ester and sulfonate S pools where the exact S species cannot be identified via operationally defined fractionation methods [1]. However, these fractionation based studies have highlighted that mobilisation of both organo-S species is important for optimisation of plant S supply [44,45].

Excluding the formation of intact AM symbioses inside the rotating microcosm cores resulted in significantly lower root colonisation for both plants analysed when compared to the static cores. While it is plausible that the AM hyphal ingrowth into organo-S enriched soil is beneficial for the overall mycorrhization levels of the host plant, it is important to note that the total amount of soil in the cores represents less than 10% of the soil available for fungal colonization. Consequently, it is possible that the rotating of the cores to prevent fungal ingrowth may not only limit the access to nutrients but could also disturb the overall mycorrhization of the plant host. While the present study has also utilized 1 µm mesh cores that do not need to be rotated, their use was only to compare ^34^S uptake (data not shown) and not mycorrhization.

Treseder [46] demonstrated that increased AM root colonisation improved both plant yield and nutrient content. The mechanism for which may include increased efficiency of nutrient mobilisation, uptake and transfer via characteristic mycorrhizal structures [47,48]. In the present study, the promoted AMF symbiosis via static cores has not only resulted in an increased percentage root colonisation but has also stimulated bacterial proliferation. Indeed, more abundant cultivable heterotrophs, sulfonate and polymeric sulfonate mobilisers were not only found in the static cores when compared to the rotated ones, but they were also more abundant in the rhizosphere of microcosms with static cores. It appeared that there is a putative positive knock-on effect from the enhanced mycorrhization through access to staic cores that had manifested itself in the bacterial community. In turn, the exclusion of fungal ingrowth into rotating cores had a more profound limiting effect. A previous study has shown that sulfonate mobilisers are abundant in the hyphosphere, putatively attaching to the hyphal surface via a type III secretion system [15]. Likewise, AMF inoculation was found to alter bacterial community dynamics that included sulfonate utilizing bacteria [49]. These recent findings, appear to support the here hypothesised positive knock-on effect.

Previous studies demonstrated that AM fungal symbiosis alters microbial community composition in the hyphosphere and rhizosphere [15,50]. Thus, it was not surprising that community fingerprint analysis in the present study of the organo-^34^S microcosms identified significant bacterial, fungal and AM fungal community shifts (presented as DCA biplots) for the static (mycorrhizal) and rotating (severed hyphae) treatments. The above hypothesized positive knock-on effect of the hyphal ingrowth in the cores may explain why bacterial rhizosphere community structures were clearly and significantly differentiated as well. Studies on bacterial diversity and abundance associated with AM hyphae have presented equivocal results in the past and the findings in this study were deemed to be a result of AM induced physico-chemical modification of rhizodeposition stimulating dominance of certain bacterial groups [51,52,53].

The amplicon-based sequencing (NGS) approach in the present study showed that Planctomycetes were stimulated greatly in both the static and rotating cores with highest numbers exceeding 34% relative abundance in the mycorrhizal (static) cores. The Proteobacteria were the second dominating phylum in the present study but unlike the Planctomycetes were equally abundant in the cores and rhizospheres. Proteobacteria groups have been identified to be S mobilisers in the past [11] and the current study is no exception when looking at the *asfA* affiliations. Within the Proteobacteria, the *Xanthomonadaceae* were most abundant in the static cores with the presence of AM hyphae and this family includes *Stenotrophomonas* which was recently isolated directly from the hyphosphere and putatively has the ability to attach to and co-migrate with AM hyphae [15]. However, while the abundance of *Xanthomonadaceae* was significantly lower in the rhizosphere, the abundance of *Xanthomonadaceae* was not significantly lower in the rotating cores over the static cores. It is possible that clades of this family can be found on various types of hyphe and not only AM fungal hyphae. Likewise, the *Comamonadaceae* appeared to be stimulated in the mycorrhizal static cores but this observation did not reach significance (*p* > 0.05). *Comamonadaceae* repeatedly associated with aromatic sulfonate desulfurization include *Variovorax* and *Polaromonas* [12,14]. While both genera were present in the cores and rhizospheres of the present study, no significant difference in relative abundance was identified.

Sulfate-ester (arylsulfatase enzyme assay) and sulfonate mobilising potential (MPN of sulfonate utilizing bacteria) was significantly increased in the static cores presumably by the stimulated AM symbiotic partnership. However, arylsulfatase activity was higher in the rhizosphere of the microcosms with static cores than within the static cores themselves. Since the soils in the cores were treated with glucose and ^34^SO_4_ to establish organo-^34^S, one could speculate that this treatment had an effect not dissimilar to a fertilizer treatment. S-fertilization has been demonstrated to decrease arylsulfatase activities [9], thus it is possible that the core soil treatment had a prolonged negative effect on the arylsulfatase activity. This is in direct contrast to the abundances of sulfonate utilizing bacteria that was highest in the static cores. Previously, the abundance of sulfonate utilizing bacteria was unaffected by fertilization [9], though the amendment of soil with biochar did increase the abundance of these bacteria [14]. Following these observations, one may speculate that within the static cores recently formed sulfonates are predominantly mobilized, while outside the cores in the rhizosphere more of the soil sulfate esters are utilized. However, the XANES spectra of the cores do not support the hypothesis of a selective utilization of sulfonates in static cores. Likewise, in an additional experiment the authors of the present study have established static and rotating cores with soil enriched with ^34^S that underwent an additional sulfatase treatment [54]. There, no significant increase in ^34^S plant uptake was identified. Therefore, it is unlikely, that a preferential mobilization of carbon bonded S over ester bonded S took place in the static cores over the duration of the experiment. While the plant ^34^S uptake was monitored over time, the remainder of the analyses were only possible after destruction of the microcosms, hence ^34^S uptake at month 3 and 6 cannot be directly compared to the remainder of the results. It is therefore possible that a shift in organo-S mobilization and utilization took place inside and outside the cores. The highlighted sulfatase activity and abundance of sulfonate utilizing bacteria is therefore only a snapshot at month 9 when most of the ^34^S from the enriched core soils was already mobilized by month 6. Further studies would be beneficial that would carry out enzymatic and microbial analyses within the first 3 months of incubation. Nevertheless, previous studies have shown improved S uptake in the presence of AMF which may be a result of the carbon rich hyphosphere selecting for organo-S mobilising bacteria with the potential to improve nutrient uptake [21,55,56]. Thus, the current study is confirming these previous observations.

Analysis of the diversity of the sulfate-ester mobilising *atsA* gene revealed differences in presence and relative abundance of taxa in the cores and rhizospheres. While *atsA* can be considered to be best studied in the Proteobacteria *Pseudomonas* and *Klebsiella* [5,6], there was no evidence of a substantial presence of *atsA* from Proteobacteria in the present study. The *atsA* clone libraries were dominated by the Acidobacteria and Verrucomicrobia. The former was the most relative abundant phylum in the rhizosphere. In contrast, the Rhodospirillales OTUs were almost exclusively found in the cores. The Planctomycetes *atsA* clones were exclusively found in the static cores. Based on NGS sequencing data, Planctomycetes were among the most abundant phyla in the microcosms overall. Interestingly, the majority of Planctomycetes 16S sequences were found in the static cores, followed by the rotating cores, while the rhizosphere harboured only a small fraction of the Planctomycetes. This phylum shows some rather unusual cell features when compared to most other bacterial phyla and is commonly detected in marine environments. Some marine isolates were identified for industrial use of their sulfatase enzymes [57]. More recently, it has also been regularly identified in soils [58] and in the present study it’s sulfatase activity may be of importance in connection to the hyphosphere and ultimately to plant-S uptake. However, up to date very limited information is available on microbial sulfatase genes in rhizo- and hyphosphere, thus further research is needed in this area.

Clear differences in presence and relative abundance of genera were observed for the aromatic sulfonate mobilising community (*asfA*). While taxonomic affiliation via BLAST resulted in most sequences to be associated with Rhizobiales and Comamonadaceae, integration into the phylogenetic tree of AsfA [11,12,13,14,15] associated 41% of all OTUs with the genus *Cupriavidus*. Most of the OTUs came from the cores and of them again the majority was found in the static cores, hence presumptively associated with AM fungal hyphae. Previous studies that assessed the diversity of *asfA* in soil, rhizosphere [11,12,13,14,15] and hyphosphere [11,12,13,14,15] did not pick up sequences related to *Cupriavidus* with the exception of one *Cupriavidus* like OTU identified in the rhizosphere of *Lolium perenne* [11,12,13,14,15]. One may speculate that *Cupriavidus* may play a particular role in sulfonate-S mineralization in the hyphosphere of mycorrhiza that are in particular associated with *Agrostis*. The genus *Cupriavidus* has been linked with improved P and N mobilisation in the past [59,60] and may play an important role in overall plant nutrition that now includes also S. Cultivation dependent data from the present study are in line with the cultivation independent results as the MPN approach for sulfonate utilizing bacteria cultivation identified highest abundances in the static cores. Other commonly retrieved sequences of AsfA clustered with *Variovorax* that dominated the rhizosphere but was also found in the rotating cores. *Variovorax* was identified as a common sulfonate mobiliser in various rhizospheres including that of *Agrostis rupestris* of the Damma glacier forefield in Switzerland [61] and the potato rhizosphere from the Netherlands [62]. Particular dominance of *Variovorax asfA* was detected in the rhizosphere of wheat in the long-term experiment Broadbalk, managed by Rothamsted Research (UK) [12].

In conclusion, this study demonstrates that an intact AM symbiosis increases plant uptake of ^34^S from organo-^34^S enriched soil at early stages of growth when S requirement is high. This may be a result of AM induced microbial community shifts leading to improved sulfate-ester and sulfonate mobilising activity. Likewise, allowing fungal ingrowth in the static cores has positively impacted mycorrhization in the microcosms and had a positive feedback onto other microbial putative activities. Therefore, fostering AM symbiosis has the potential to improve plant S uptake and reduce the requirement for unsustainable inorganic S fertilisation practices, while in turn limiting AM access to resources has a potentially negative impact that extends to the plant host and to the microbes inhabiting the hyphosphere and rhizosphere.

## 4. Materials and Methods

### 4.1. Site Description

The soil used to create the experiments was obtained from a long-term field trial (“Cowlands”) at Teagasc, Johnstown Castle, Wexford, Ireland (52°16′ N, 6^°^30′ W, 30 m above sea level). The soil type is a poorly drained gley soil classified as Mollic Histic Stagnosol (WRB 2006) with 11% of organic matter, a pH of 6 and a loamy topsoil with 18% of clay content. The soil has not received P or S fertiliser since 1968 and has not been ploughed since 1970 (P0-0, site A5). Swards are mixtures of *Lolium perenne*, *Dactylis glomerata* and various meadow grass species [63].

### 4.2. Stable Isotope Enriched Soil

The most common form of elemental S in nature is ^32^S (95.02%) and stable isotope ^34^S represents 4.21% of S existing naturally in soil. For this study, S uptake and incorporation into plant biomass was undertaken using the ratio of ^34^S to ^32^S (δ^34^S values) [64].

To enrich the soil with organo-^34^S, 20 g soil aliquots were treated with ^34^S-SO_4_ (20 mg/kg soil) and glucose (6 g/kg) to stimulate S immobilisation at 80% water holding capacity (WHC) and optimise the amount of S in the C-bonded fraction [65]. The soil was incubated for 6 weeks at 25 °C and aerated fortnightly. After 6 weeks, the 20 g soil was thoroughly mixed with 20 mL of 0.01 M CaCl_2_ for 30 min on a Intelli-Mixer RM-2 (Elmi Tech Ltd., Brea, Latvia) and subjected to centrifugation (Allegra X-22R, Beckman Coulter, Brea, CA, USA) at 3900 rcf for 20 min and to remove inorganic S [65]. This process was repeated two additional times before the supernatant was discarded and the residual material was dried, ground and brought to 60% WHC. To counteract the loss of essential nutrients, the enriched fraction was supplemented with Hoagland’s solution (S free, 50%) [66]. The ^34^S enriched soil samples were transferred into nylon mesh cores (up to 6 g per core) with windows of either 35 µm or 1 µm adapting an experimental setup described previously [40].

### 4.3. Core Design and Constructing the Soil Microcosms

Cores were constructed adapted from a design published by Johnson and colleagues [40] using acrylonitrile butadiene styrene (ABS) water pipe (150 mm height, 18 mm diameter). Two windows were cut into each core, the size of which constituted 90% of the surface area of the core (Appendix A). Nylon mesh (Plastok Associates Ltd., Birkenhead, UK) with 35 µm pores was used to cover the windows of the core and 1 µm nylon mesh was used to cover the base. The mesh was attached using Tensol No. 12 adhesive (Evode Speciality Adhesives Ltd., Leicester, UK) and the completed cores were heated (24 h, 80 °C) to harden the adhesive (cured). The 35 µm mesh is sufficiently large to allow mycorrhizal hyphae to colonise the cores while preventing root access. A permeable base of 1 µm allows free drainage of water through the cores and minimises hyphal access. This set of cores was used to analyse the role of AM in uptake of organo-^34^S where the plants are cultivated outside the cores and AM hyphae are able to grow inside the cores enriched with organo-^34^S.

### 4.4. Organo-^34^S Enriched Soil Microcosms for Cultivation of Agrostis stolonifera and Plantago lanceolata

Microcosm systems were constructed from cuttings of Plexiglas (12 × 11 × 2.5 cm, chemically welded with Ethylene Dichloride) and filled with 230 g of a sand (Glenview Natural Stone, Ireland) and soil (Teagasc, Johnstown Castle) mixture (one part sand, one part soil *w/w*). These were planted with *A. stolonifera* and *P. lanceolata*, respectively and inoculated with 2 g of *Rhizophagus irregularis* (purchased as *Glomus intraradices;* Symbivit, Symbion, Czech Republic). For this experiment, two mesh cores, containing 6 g of organo-^34^S stable isotope enriched soil, were inserted into each soil microcosm (two cores per microcosm were chosen to minimise the distance of plant roots to the cores, see also Appendix A). The cores were kept plant-free and served as the organo-^34^S source in this experiment. The organo-^34^S enriched cores were either rotated twice weekly to sever AM hyphae (rotating) or undisturbed (static) to allow AM colonisation. Each treatment was carried out in replicates of six (12 *Agrostis* microcosms, 6 with rotating cores and 6 with static cores; 12 *Plantago* microcosms, 6 with rotating cores and 6 with static cores). Plan shoots were repeatedly cut for ^34^S analysis, while cores and rhizospheres were sampled destructively at the end of the experiments. The microcosms were grown in an A1000 Adaptis plant growth chamber (Conviron, Germany) with day-night temperature of 25–15 °C, respectively, 12 h day, 70% RH, and 320 µmoles m^−2^ s^−2^ photosynthetic active radiation (PAR). The systems were watered with dH_2_O three times a week and once fortnightly supplemented with half times S free Hoagland’s solution [66].

### 4.5. Determination of ^34^S Uptake into Shoots of A. stolonifera and P. lanceolata

Determination of ^34^S uptake was achieved using Elemental Analysis—Isotope Ratio Mass Spectrometry (EA-IRMS) undertaken by Iso-Analytical (Cheshire, UK) as outlined in the Appendix A. The aboveground biomass was destructively harvested by cutting off shoots at 5 cm above the ground to analyse the S flow from labelled organo-^34^S in the mesh cores to its incorporation into plant tissue. *P. lanceolata* was measured at 3 and 6 months (3 replicates per treatment). *A. stolonifera* was measured at 3, 6 and 9 months (6 replicates) (see also analysis flow chart, Appendix A). The plant biomass was dried at 80 °C for 5 d, ground to a fine powder and a 0.5 g aliquot was subjected for EA-IRMS. Additionally, at 9 months one of the two cores in each *A. stolonifera* microcosm was re-planted into a new microcosm instead of being destructively harvested (second round of microcosms with one core). The second generation of microcosms were established to ascertain long term organo-^34^S uptake after 12 months of incubation in total (6 replicates).

### 4.6. S K-Edge X-ray Absorption near Edge Spectroscopy

XANES analysis was used to identify the chemical oxidation states of S, reduced (sulfides, thiols), intermediate (sulfoxides, sulfonates) and oxidised (sulfate esters, sulfates) as described previously [30]. In the present case, soil subsamples from two static and two rotating core experiments of *A. stolonifera* were compared. Soil samples were dried at 105 °C for 12 h and were ground to a fine powder using a pestle and mortar. SO_4_^2−^ was extracted from each of the samples with 0.01 M CaCl_2_ using the same approach highlighted above in Section 4.2. The soil was freeze dried in a Labconco FreeZone™ 4.5 L Freeze dry system (Labconco, Kansas City, MO, USA). XANES was carried out on the KMC-1 beam line at BESSY II (Helmholtz-Centre Berlin for Materials and Energy, Berlin, Germany, for further details see Appendix A). The KMC-1 is a soft X-ray double crystal monochromator beamline for the energy range 2–12 keV. The photon energy of sulfate is the same as for sulfate esters. Since free sulfate was removed from the samples prior to testing, the corresponding peaks are therefore exclusively from sulfate esters.

### 4.7. Percentage Root Colonisation

At the time of harvest, each plant type and treatment was examined for root colonisation by AM fungi using a modified version of the grid line intersect method [67] as described in the Appendix A (analysis flow chart, Appendix A). Arbuscular colonization (AC) and vesicular colonization (VC), respectively, were calculated by dividing their respective counts by the total number of intersections examined. Hyphal colonization (HC) was calculated as a proportion of the positive intersections. For each organo-^34^S stable isotope soil microcosm, 3 replicates of 100 fields of view per treatment, per plant were examined to calculate AM fungal colonisation.

### 4.8. Extraction and Quantification of Bacteria from AM Hyphae

To harvest the microcosm cores and plant roots of the static and rotating treatments of both plant types (six cores and plant roots per treatment), up to 1 g of hyphosphere material (hyphae with adhering soil, handpicked with fine forceps under a stereomicroscope) was collected from each core and 1 g of rhizosphere (roots with adhering soil) was analysed in parallel. The hyphosphere and rhizosphere bacteria were extracted into 10 mL of sterile saline solution (0.85%) and rotated at 75 rpm on an Elmi Intelli-Mixer RM-2 (Elmi Tech Ltd., Latvia) (30 min, 4 °C). A 0.1 mL aliquot of the resulting suspension was used for bacterial community quantification (cultivation dependent). The remainder of the suspensions were centrifuged at 4500 rpm (4 °C, 20 min) and the obtained pellets were immediately frozen (−18 °C) for subsequent cultivation independent analysis (see Section 4.9, Section 4.11 and Section 4.12 below).

The aliquot of bacterial suspension from above was used for tenfold serial dilutions in saline (0.85% *w/v*). Most probable number (MPN) analysis was undertaken in agar-free R2A [68], MM2TS and MM2LS ([14]; Appendix A)) to enumerate the cultivable heterotrophic, sulfonate and polymeric sulfonate mobilising communities, respectively. A 20 µL aliquot from each dilution (10^1^–10^7^) was added to 200 µL of either R2A MM2TS or MM2LS in 96 well microtiter plates. Following 2 weeks of growth in an Innova Incubator Shaker Series (New Brunswick Scientific, Edison, NJ; 75 rpm, 25 °C), the OD_590_ was recorded using an ELX808IU spectrophotometer (3 min shaking at level 3; Bio Tek Instruments Inc., Winooski, VT) in order to determine the MPN number which in turn was used to obtain an MPN g^−1^ value [69].

### 4.9. Community Fingerprinting

The frozen pellets from the organo-^34^S microcosms of the bacterial extractions were used for community DNA extractions using the UltraClean Soil DNA extraction kit from MoBio as described by the manufacturer (Carlsbad, CA, USA). Subsequent Denaturing Gradient Gel Electrophoresis (DGGE) was carried out for the bacterial 16S rRNA, 18S AM fungal and ITS saprotrophic fungal communities as described previously [70]. PCR and DGGE conditions as well as a list of the selected primers for PCR-DGGE are available in the Appendix A and Appendix A.

### 4.10. Arylsulfatase Activity

A colorimetric method of assaying soil arylsulfatase activity was used to quantify arylsulfate ester cleaving enzymes in 1 g of soil with *p*-nitrophenyl sulfate as the substrate [71]. Arylsulfatase activity was analysed for hyphosphere (inside cores) and rhizosphere, static and rotating treatments. Further details are provided in Appendix A.

### 4.11. Diversity of asfA and atsA Gene

Diversity of the bacteria with the capacity to mobilize sulfur from sulfonates and sulfate esters from the *A. stolonifera* microcosms were analysed via their marker genes *asfA* and *atsA*, respectively. Amplicons of the corresponding genes were used to generate respective clone libraries of the hyphosphere and rhizosphere from the static microcosms as well as the hyphosphere and rhizosphere from the rotating microcoms.

The *asfA* gene was amplified with primers asfAF1all [15] and asfBtoA [11] (Appendix A). A touchdown PCR was carried out under the following conditions: initial denaturation at 98 °C for 3 min, 10 cycles of 98 °C denaturation (10 s), 65–55 °C touchdown (15 s), 68 °C extension (40 s), plus 25 further cycles at 58 °C annealing. Final extension was carried out at 68 °C for 3 min. PCR was undertaken in 25 µL reactions containing; 1 X Terra PCR Direct Buffer (2 mM MgCl_2_), 0.2 mM dNTP mix, 0.4 µmol of each primer, and 0.7 U of Terra PCR Direct Polymerase (ClonTech Europe, Saint-Germain-en-Laye, France).

The *atsA* gene was amplified using the AtsA-F1 AtsA-R1 primer pair [9] (Appendix A). PCR conditions were as follows: initial denaturation at 95 °C for 3 min, 40 cycles of 95 °C denaturation (60 s), 48 °C annealing (60 s), and 72 °C extension (60 s). Final extension was carried out at 72 °C for 10 min. The PCR was undertaken using a Kapa 2G Robust PCR kit (Kapa Biosystems, Woburn, MA, USA) in 25 µL reactions containing; 1× Buffer A, 1× Enhancer, 5% DMSO (Sigma-Aldrich, St. Louis, MO, USA), 2 mM MgCl_2_, 0.2 mM dNTP, 0.5 µM of each primer and 0.5 U of the Kapa Robust polymerase.

The *asfA* and *atsA* PCR products were purified, quantified, and ligated into the cloning vector pJET1.2/blunt (CloneJet, Thermo Scientific, Waltham, MA, USA) and pGEM^®^-T vector kit (Promega, Madison, WI, USA), respectively. The ligation and transformation steps were undertaken as per manual instructions. The ligation reactions were transformed into *E. coli* DH5α. In order to ascertain taxonomic diversity of recombinant plasmids containing an insert of the correct size, Restriction Fragment Length Polymorphism (RFLP) analysis was carried out on PCR amplicons using the restriction enzymes *AluI* and *RsaI* (5 U per reaction; Thermo Scientific) for 4.5 h at 37 °C. The digested DNA was run on a 2% agarose gel at 100 V for 40 min. Clones with a similar restriction pattern were classified as a genotype (OTU) using Phoretix 1D (Nonlinear Dynamics, Newcastle upon Tyne, UK). Library coverage was calculated as one, subtracted by the ratio of the total number of OTUs with single clones to the total number of clones multiplied by hundred [34]. Unique genotypes with more than one representative were re-amplified and the purified PCR product was used for sequence identification (GATC Biotech, Konstanz, Germany). The sequences obtained were subjected to gene comparison using BLAST [31]. Sequences of *asfA* were imported into arb (Version 5.2) [72], translated into proteins and integrated into an established *asfA* phylogenetic trees [32]. Nucleic acids were deposited in EBI’s nucleic acid archive under the accession numbers MT309545-MT309559 & MT380723 (*atsA*) and MT309560-MT309575 & MT372478-MT372479 (*asfA*).

### 4.12. Amplicon Sequencing of 16S sRNA Gene Fragments (NGS)

For Illumina MiSeq 16S library preparation, community DNA from *A. stolonifera* (after 9 months of incubation) was amplified (cores and rhizosphere, static and rotating microcosms, replicates of six) using the primer pair 16SF and 16SR to target the V3 and V4 region yielding a 460 bp amplicon (Illumina Metagenomic Sequencing Library Preparation Guide; Appendix A). PCRs were undertaken in 25 µL reactions with 0.5 U of Kapa HiFi Taq, 1× PCR buffer with 1.5 mM MgCl_2_, 0.2 mM dNTPs each (all Kapa Enzymes, Woburn, MA, USA), and 0.4 µM of each primer. A touchdown PCR protocol was used with the following cycling conditions: initial denaturation at 98 °C for 5 min, 20 cycles of 98 °C denaturation (45 s), 68–58 °C touchdown (60 s), 72 °C extension (60 s), plus 20 further cycles with an annealing temperature at 58 °C. The PCR product was purified using the GenElute PCR purification kit (Sigma-Aldrich, St. Louis, MO, USA). The indexing PCR was carried out to attach the dual indices and Illumina sequencing adapters using the Nextera XT Index Kit (Illumina, San Diego, CA, USA) in accordance with the manufacturer’s instructions with a modified annealing temperature of 63 °C. The indexing PCR product was purified (as before) and quantified using the Qubit dsDNA HS Assay Kit (Life Technologies, Carlsbad, CA, USA) on a Qubit 2.0 Fluorometer (Life Technologies, Carlsbad, CA, USA). The DNA concentrations of each sample were adjusted to 4 nM in 10 mM Tris pH 8.5 and 5 µL was used to mix aliquots for pooling libraries with unique indices. Amplicons were directly sequenced on an Illumina MiSeq NGS platform (Illumina, San Diego, CA, USA) in line with protocols at the Teagasc, Moorepark sequencing centre.

### 4.13. Data Analysis

Analysis of variance (ANOVA) of EA-IRMS data, percentage root colonisation, and MPN data was carried out using IBM SPSS statistics (Version 22.0; IBM, Armonk, NY, USA). Data were tested for normality and homogeneity of variance (Shapiro–Wilk’s and Levene’s tests). Data that were not normally distributed were transformed logarithmically, analysed by ANOVA and then back transformed. Where normality was confirmed, a Tukey’s HSD post hoc test was applied to assess any significant differences (*p* < 0.05). Where homogeneity of variance was not achieved, the Games-Howell test was used instead. Pairwise comparison of static versus rotating core as well as core versus rhizosphere sequencing data was carried out using the t-test function. DGGE fingerprinting gels were digitalised and band patterns analysed with the software package Phoretix 1D (Nonlinear Dynamics). Cluster analysis using UPGMA was carried out and obtained band pattern matrixes were exported for Detrended Correspondence Analysis (DCA) and permutation tests (Monte-Carlo with 9999 replicates) as described previously [32]. Error ranges are reported as standard deviation throughout the manuscript.

The following steps were undertaken to analyse the Illumina NGS data; raw Illumina paired-end sequence reads were merged using Flash [73] and quality checked using the split libraries script in QIIME [74]. Quality checking is two phase; for joining reads a min overlap of 10 bp was expected; then for joined reads anything less than 150 bp and quality of 19 was removed. Reads were clustered into OTUs with a similarity cut off at 97% and chimeras were removed with the 64-bit version of USEARCH [75]. Subsequently, OTUs were aligned and a phylogenetic tree was generated within QIIME [74]. Alpha and beta diversity analysis was also implemented within QIIME [74]. Alpha diversity was determined using the Shannon and Simpson diversity indices and the *Chao*1 richness estimator. Beta diversity was analysed using PCoA (weighted Unifrac) undertaken with EMPeror [76]. Taxonomical assignments were reached using the 16S-specific SILVA database (Version 106). Sequences were deposited in the EBI’s nucleic acid archive with the accession numbers (ERS4414312-ERS4414324).

## Figures and Tables

**Figure 1 plants-11-03050-f001:**
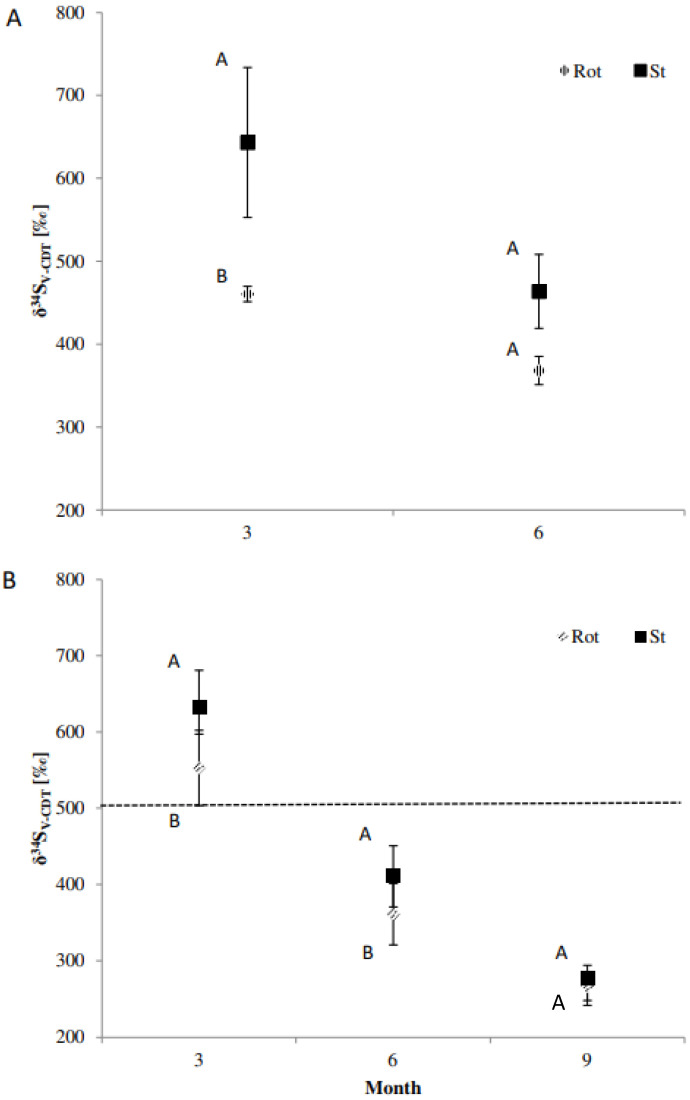
AM uptake of ^34^S from organo-^34^S following 3, 6 and 9 months of growth. (**A**) *= Plantago lanceolata*, (**B**) *= Agrostis stolonifera,* rotating (Rot) *=* severed hyphae, and static (St) = mycorrhizal. *P. lanceolata* was measured at 3 and 6 months, only (3 replicates per treatment). *A. stolonifera* was measured at 3, 6 and 9 months (all 6 replicates were analysed). Letters (A,B) indicate significant differences. The level of background (leaching) ^34^S uptake was measured at 3 months for *A. stolonifera* and is highlighted (dashed horizontal line).

**Figure 2 plants-11-03050-f002:**
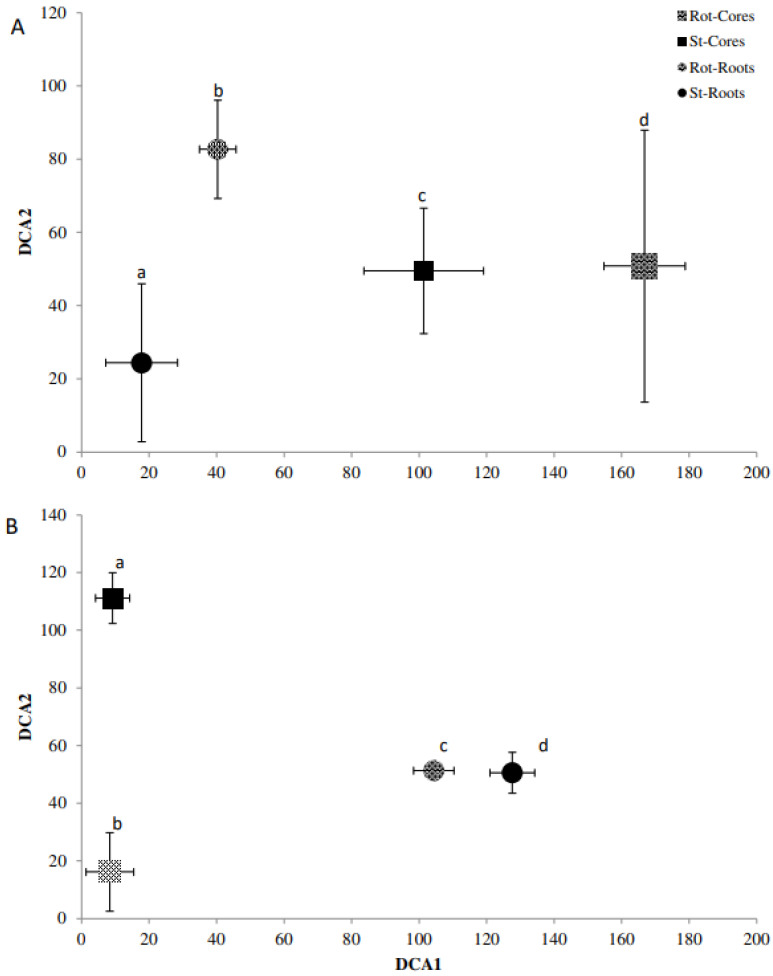
Detrended correspondence analysis (DCA) bi-plot of the 16S rRNA community fingerprint for *Agrostis stolonifera* (**A**) and *Plantago lanceolata* (**B**). DCA1 = Axis 1, DCA2 = Axis 2. Treatments; roots = circle, cores = square, static (St) = solid, rotating (Rot) = pattern. Letters (a–d) indicate significant differences; error bars represent standard deviation among replicates.

**Figure 3 plants-11-03050-f003:**
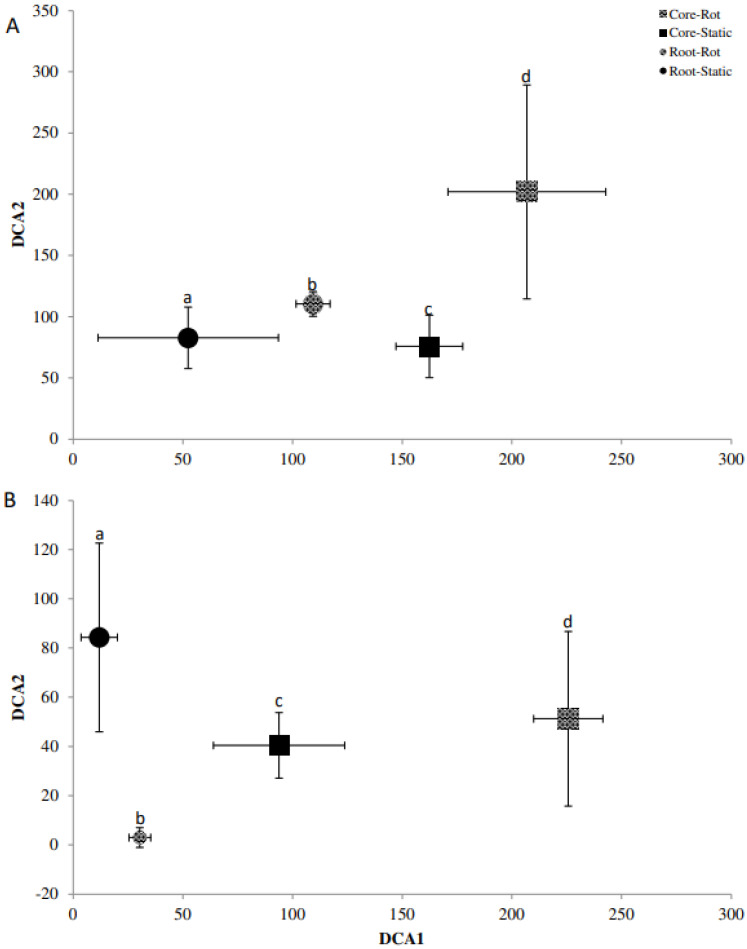
Detrended correspondence analysis (DCA) bi-plot of the 18S AM fungal community fingerprint for *Agrostis stolonifera* (**A**) and *Plantago lanceolata* (**B**). DCA1 = Axis 1, DCA2 = Axis 2. Treatments; roots = circle, cores = square, static (Static) = solid, rotating (Rot) = pattern. Letters (a–d) indicate significant differences; error bars represent standard deviation among replicates.

**Figure 4 plants-11-03050-f004:**
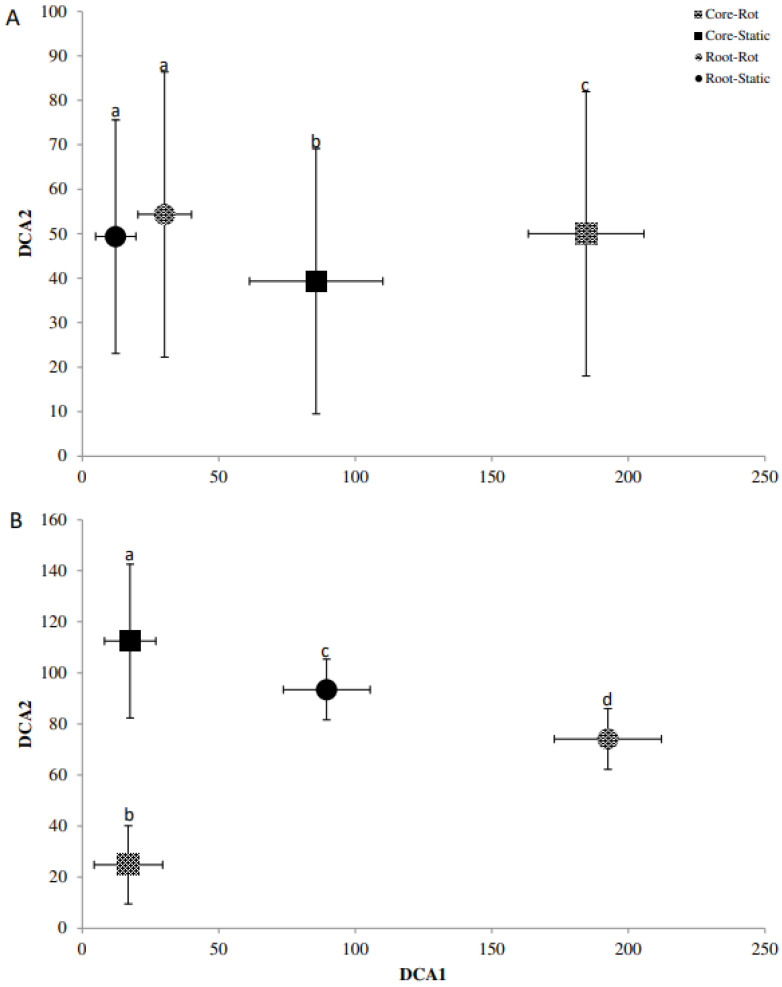
Detrended correspondence analysis (DCA) bi-plot of the fungal ITS community fingerprint for *Agrostis stolonifera* (**A**) and *Plantago lanceolata* (**B**). DCA1 = Axis 1, DCA2 = Axis 2. Treatments; roots = circle, cores = square, static (Static) = solid, rotating (Rot) = pattern. Letters (a–d) indicate significant differences; error bars represent standard deviation among replicates.

**Figure 5 plants-11-03050-f005:**
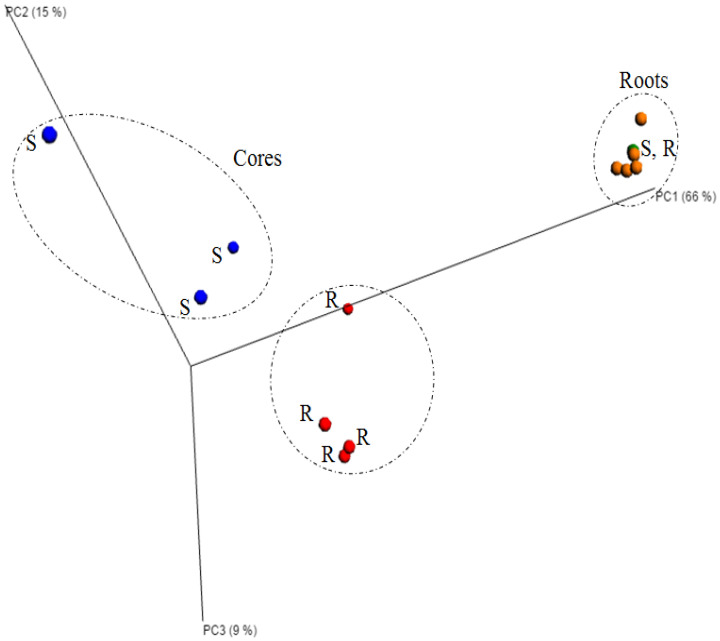
Principal Ccoordinates Analysis (PCoA) of bacterial community sequences based on 16S rRNA amplicons from hyphosphere cores (red (static, S) and blue (rotating, R)) and rhizosphere roots (green (static, S) and orange (rotating, R)) of *Agrostis stolonifera*. Static = mycorrhizal and rotating = severed mycorrhizal hyphae treatments. PCoA was calculated using an un-weighted Unifrac distance matrix and visualised with EMPeror.

**Table 1 plants-11-03050-t001:** Arbuscular mycorrhizal colonisation and abundance of cultivable bacteria (most probable number, MPN) from static and rotating cores as well as rhizospheres from microcosms with static and rotating cores (*n* = 6). Letters (a–d) indicate significant differences in mycorrhization and bacterial abundances across cores and rhizospheres (static and rotating).

	*Agrostis stolonifera*				*Plantago lanceolata*			
	**Cores**		**Rhizosphere**		**Cores**		**Rhizosphere**	
	**Rotating**	**Static**	**Rotating**	**Static**	**Rotating**	**Static**	**Rotating**	**Static**
**Mycorrhization [%]**								
Arbuscular Colonisation			43.9 a	73.3 b			48.1 a	82.1 b
Vesicular Colonisation			24.2 a	40.3 b			29.5 a	47.7 b
Hyphal Colonisation			38.9 a	66.4 b			44.7 a	75.4 b
**Bacterial Abundance**								
Heterotrophs ^1^ [MPN/g 10^7^]	2.0 a	11.6 b	1.8 a	11.5 b	2.6 a	16.9 b	1.2 a	15.8 b
Arylsulfonate utilizers ^2^ [MPN/g 10^4^]	152.0 a	1930.0 b	2.9 c	14.1 d	107.0 a	1630.0 b	3.9 c	1200.0 b
Alcylsulfonate utilizers ^3^ [MPN/g 10^3^]	6.13 a	124 b	5.7 a	136 b	97.4 a	159.0 a	14.1 b	149.0 a

^1^ Reasoner 2 medium without agar. ^2^ Minimal medium with toluenesulfonate as sole sulfur source. ^3^ Minimal medium with lignosulfonate as sole sulfur source.

**Table 2 plants-11-03050-t002:** Taxonomic assignment of representative desulfonating operational taxonomic units (OTUs) derived from clonal *asfA* DNA sequence analysis from static hyphosphere cores (CS), rotating hyphosphere cores (CR), rhizosphere roots from microcosms with static cores (RS) and rhizosphere roots from microcosms with rotating cores (RR).

OTU ID	Affiliation	Abundance per Treatment				Similarity [%] DNA/Protein
		CS	CR	RS	RR	
RR2	α-Proteobacteria				8	70/84
RR8	α-Proteobacteria				4	71/68
RS34	α-Proteobacteria			3		70/68
CS37	Rhizobiales	5	11	2	6	72/66
CR11	Rhizobiales		6			72/66
CS16	Rhizobiales	29	10	5	2	72/66
RR6	Rhizobiales				7	71/74
RS6	Comamonadaceae			13		72/76
RR34	Comamonadaceae				3	71/71
RS4	Comamonadaceae			4		78/92
RS40	Comamonadaceae			6		82/94
CR38	*Stenotrophomonas*		8			65/88
CS12	*Paraburkholderia*	3				99/98
RR29	*Variovorax*				5	81/95
CR8	*Variovorax*		4			88/99
CR13	*Variovorax*		4			83/92
RR22	*Variovorax*				4	82/86
RS20	*Variovorax*					82/86

**Table 3 plants-11-03050-t003:** Taxonomic assignment of representative desulfonating Operational Taxonomic Units (OTUs) derived from clonal *atsA* nucleic acid sequence analysis from static hyphosphere cores (CS), rotating hyphosphere cores (CR), rhizosphere roots from microcosms with static cores (RS) and rhizosphere roots from microcosms with rotating cores (RR).

OTU ID	Affiliation	Abundance per Treatment				Overall Abundance [%]	Similarity [%] Protein
		CS	CR	RS	RR		
1	Bacteroidetes		2	6	12	12.5	64
7	Acidobacteria	1	3		7	7.5	68
16	Acidobacteria	3				2	72
19	Acidobacteria	1		5	6	8	77
20	Acidobacteria				6	4	97
36	Acidobacteria			3		2	79
33	Acidobacteria			6		4	57
2	*Chthoniobacter*		2	1	2	3	73
14	Verrucomicrobia		4			3	68
3	Verrucomicrobia	3	2	2	1	5.5	65
10	Verrucomicrobia		2		2	3	68
23	Verrucomicrobia	3			1	2.5	57
18	Planctomycetes	6				4	59
4	Rhodospirillales (order)		6			4	51
5	Rhodospirillales (order)	7	2		1	6	54
30	Eukaryotes, Rotifera			6		4	73

**Table 4 plants-11-03050-t004:** Relative abundance (%) of 35 most abundant bacterial families in *Agrostis stolonifera* microcosms (16S rRNA gene fragment amplicon libraries) from static hyphosphere cores (CS), rotating hyphosphere cores (CR), rhizosphere roots from microcosms with static cores (RS) and rhizosphere roots from microcosms with rotating cores (RR); bold numbers indicate significant differences between cores and rhizosphere (C, R). Letters A–B indicate significant difference between static and rotating cores (CS, and CR, *n* ≥ 3).

Phyum	Family	CR	CS	RR	RS
Acidobacteria	Acidobactertiaceae	**0.24**	**0.18**	**4.34**	**4.38**
	Holophagaceae	**7.48**	**2.59**	**0.06**	**0.02**
	Bryobacter	**0.40**	**0.22**	**1.10**	**0.84**
Actinobacteria	TM214	**0.06**	**0.08**	**0.79**	**0.89**
	Mycobacteriaceae	**0.00**	**0.00**	**0.89**	**0.94**
	Acidothermaceae	**0.00**	**0.00**	**1.31**	**1.90**
	Micromonosporaceae	9.25	1.21	0.19	0.20
Bacteroidetes	Porphyromonadaceae	**1.60**	**0.09**	**0.01**	**0.00**
	Cytophagaceae	1.05	0.22	0.40	0.37
	Chitinophagaceae	0.31	0.66	0.67	1.16
Firmicutes	Bacillaceae	**1.90**	**1.92**	**0.49**	**0.69**
Gemmatimonadetes	Gemmatimonadaceae	**0.68 A**	**1.55 B**	**2.32**	**2.19**
Planctomycetes	Planctomycetaceae	4.13	2.95	2.00	1.98
Proteobacteria	Caulobacteraceae	**0.37**	**0.35**	**0.19**	**0.26**
	Bradyrhizobiaceae	**0.62**	**0.49**	**2.45**	**2.97**
	Hyphomicrobiaceae	**0.37**	**0.34**	**0.91**	**1.14**
	Xanthobacteraceae	**0.25**	**0.26**	**3.27**	**3.61**
	Acetobacteraceae	0.35	0.06	0.38	0.42
	DA111	**0.04 A**	**0.01 B**	**0.47**	**0.53**
	Rhodospirillaceae	1.02	0.38	0.36	0.37
	Sphingomonadaceae	0.12 A	0.46 B	0.40	0.62
	Alcaligenaceae	**0.05**	**0.07**	**0.76**	**1.15**
	Burkholderiaceae	**1.26**	**1.10**	**0.09**	**0.04**
	Comamonadaceae	1.60	3.03	1.21	1.03
	Oxalobacteraceae	**0.56**	**0.27**	**0.10**	**0.10**
	Nitrosomonadaceae	**0.27**	**0.12**	**1.64**	**1.51**
	Rhodocyclaceae	**2.04**	**0.95**	**0.29**	**0.35**
	Cystobacterineae	**0.34**	**0.07**	**0.50**	**0.47**
	Nannocystineae	**0.43**	**0.24**	**1.79**	**1.62**
	Sorangiineae	**0.21 A**	**0.03 B**	**0.84**	**0.66**
	Sinobacteraceae	**0.07**	**0.05**	**1.10**	**1.02**
	Xanthomonadaceae	**9.57**	**16.05**	**0.96**	**0.86**
Verrucomicrobia	Chthoniobacteraceae	0.18	0.31	0.21	0.28
	DA101	**0.02**	**0.01**	**6.01**	**6.02**
	Xiphinematobacteraceae	**0.01**	**0.01**	**0.65**	**0.87**

## Data Availability

Nucleic acids were deposited in EBI’s nucleic acid archive under the accession numbers MT309545-MT309559 & MT380723 (*atsA*) and MT309560-MT309575 & MT372478-MT372479 (*asfA*) and ERS4414312-ERS4414324 (16S amplicon sequences).

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
