# Peer review of "Arbuscular Mycorrhiza Support Plant Sulfur Supply through Organosulfur Mobilizing Bacteria in the Hypho- and Rhizosphere"

_plants, 2022, doi:10.3390/plants11223050_

Round 1
Reviewer 1 Report
This publication presents interesting results related to the improvement of plant sulfur supply via the interaction of arbuscular mycorrhizal fungi and organosulfur mobilizing bacteria in the hypho- and rhizosphere. The adopted approach in this study was very consistent since it could provide more insights into the understanding of the complex interactions among the different components of the plant microbiome and their role in improving plant nutrition.
The manuscript was well introduced, and the authors adopted convincing methods with a reflected discussion of the different obtained results. However, the manuscript needs minor revisions to be suitable for publication in Plants.
General comment
The English of this manuscript needs minor improvements.
Other comments
- Abstract
Keywords: please add “arbuscular mycorrhizae” as a keyword and italicize species' scientific names.
- Introduction
L30: please delete “to”.
- Results:
L95: please provide the significance of “DM” at the first appearance in the text.
L122-117: Please include this in the discussion section.
L127 and 128: please provide the significance of “AC, VC and HC” at the first appearance in the text.
L129: please delete the space between the value and the %. Please check it in the whole manuscript.
L133: “Letters (a-d) indicate significant differences”, please be more specific.
Figure 2 should be placed after its description.
L162: please provide the significance of “DGGE” at the first appearance in the text.
L186: please change “11.8 μg (± 1.90)” to “11.8 (± 1.90) μg” please correct throughout the manuscript.
L195 and 196: “12 in static cores, 10 in rotating cores, 12 in the rhizosphere of the static and 15 in rhizosphere of the rotating treatments” basically, the total should be 49 and not 43, please check.
L200: the rest of the last sentence is in L211, please correct it.
L257 and 258: please add “respectively” just after the values between parentheses.
- Discussion:
L314-316: please correct the citation form.
L325: please change “been associated with fungal biomass previously” to “been previously associated with fungal biomass”. The same in L368 and L375.
L336: please delete “be”.
L383: please do not italicize the citation.
L397: please change “speculate” to “speculates”.
Please discuss the results of DCA.
- M&M
L468: please write “materials” without “s”.
L482: please provide the significance of “WHC” and change L489 according to.
L497: “(supplementary Figure S1)”, please check, it is not the right figure and please check the others too.
L633: please delete the additional space.
Author Response
General comment
The English of this manuscript needs minor improvements.
We have now gone through the entire manuscript and have carried out minor language editing as suggested and hope that this is now adequate.
Other comments
- Abstract
Keywords: please add “arbuscular mycorrhizae” as a keyword and italicize species' scientific names.
We have made these changes as requested. L27 (all line referrals are based on the track change version of the manuscript).
- Introduction
L30: please delete “to”.
We have made these changes as requested. L30
- Results:
L95: please provide the significance of “DM” at the first appearance in the text.
We have now written out DM as dry matter. L95
L122-117: Please include this in the discussion section.
This section is a figure legend for the stable isotope analysis. We have discussed these findings in the discussion section in the last paragraph and around line 545 as well as around line 406 but have now extended this further as seen for lines 394.
L127 and 128: please provide the significance of “AC, VC and HC” at the first appearance in the text.
We have now provided the definitions of AC, VC and HC as suggested. L140-1
L129: please delete the space between the value and the %. Please check it in the whole manuscript.
We have now corrected this throughout the manuscript, thanks for highlighting this.
L133: “Letters (a-d) indicate significant differences”, please be more specific.
We have now specified where we analysed significant differences and hope this is now sufficient. L146-7
Figure 2 should be placed after its description.
We have now relocated Figure 2 as requested.
L162: please provide the significance of “DGGE” at the first appearance in the text.
We have now spelled out DGGE as requested. L171
L186: please change “11.8 μg (± 1.90)” to “11.8 (± 1.90) μg” please correct throughout the manuscript.
Thanks for highlighting, we have now made these changes throughout the manuscript.
L195 and 196: “12 in static cores, 10 in rotating cores, 12 in the rhizosphere of the static and 15 in rhizosphere of the rotating treatments” basically, the total should be 49 and not 43, please check.
These numbers can not be added as some OTUs occurred in more than one type of core or rhizosphere. Indeed, 2 OTUs occurred in all treatments.
L200: the rest of the last sentence is in L211, please correct it.
We have now moved the text section ahead of the figures in order to avoid the large gap that seemed to have appeared due to formatting issues.
L257 and 258: please add “respectively” just after the values between parentheses.
We have now made the requested correction in the first paragraph of section 2.8.
- Discussion:
L314-316: please correct the citation form.
Thanks for highlighting, we overlooked to change these citations and have now corrected this omission. L400-2
L325: please change “been associated with fungal biomass previously” to “been previously associated with fungal biomass”. The same in L368 and L375.
We have corrected the first instance as suggested but haven’t found the other two occasions mentioned above. L411
L336: please delete “be”.
We have now corrected this section as suggested.
L383: please do not italicize the citation.
This has now been corrected. L423
L397: please change “speculate” to “speculates”.
We think that in “one may speculate” is correct on this occasion.
Please discuss the results of DCA.
These results have been discussed already at lines around lines 370 but now we have made it clearer that we mean the outcome of the DCA biplots. L446-7
- M&M
L468: please write “materials” without “s”.
We are following the template provided that uses “Materials and Methods”. Therefore, we have not made the suggested change.
L482: please provide the significance of “WHC” and change L489 according to.
This has now been corrected. L569
L497: “(supplementary Figure S1)”, please check, it is not the right figure and please check the others too.
Thanks for highlighting, we had to change the order of the suppl. figure numbers due to the move of the materials and methods and must have missed the change on this occasion. L585 and L780-814
L633: please delete the additional space.
Done, L721

Reviewer 2 Report
The work by Gahan and collaborators entitled ¨Arbuscular mycorrhiza...¨ attempts to understand the role of bacteria associated with mycorrhizae and how such an important nutrient such as sulfur can be mobilized in such a symbiosis in a created environment such as a microcosm.
The article is a very good work, complete and with enough results to be published in the journal Plants. It seems to me that it could be of interest to a wide audience interested in the interactions between bacteria, mycorrhizal fungi and plants, with a wide potential in agriculture.
It seems to me that the article presents good statistical analysis and an excellent discussion.
I only have one comment that would help improve the article, and that is that the figures from one to five are too big, and the letters are blurred. I don't know if increasing the definition or making them smaller would help to have a better resolution when reading the article.
Author Response
It seems to me that the article presents good statistical analysis and an excellent discussion.
Thanks for your positive feedback.
I only have one comment that would help improve the article, and that is that the figures from one to five are too big, and the letters are blurred. I don't know if increasing the definition or making them smaller would help to have a better resolution when reading the article.
Thanks for bringing this up. We wonder why the letters occur blurred in your version as they don’t seem to be blurred in our version. There may be an issue with the processing of the manuscript as the figures were reformatted after the file upload. We will highlight this at the final production process so that the figures are all high quality. We have now reduced the figure sizes so that they line up with the text as requested.
